# Identifying the top research priorities in medically not yet explained symptoms (MNYES): a James Lind Alliance priority setting partnership

Christina Maria van der Feltz-Cornelis [ID],[1,2,3,4,5] Jennifer Sweetman [ID],[1] Mark Edwards,[6,7] Nicholas Gall,[8] Jennifer Gilligan,[9] Stephanie Hayle,[10] Arvind Kaul,[6,7] Andrew Stephen Moriarty [ID],[1,2] Petros Perros,[11] James Sampford,[9] Natalie Smith,[1] Iman Elfeddali,[12,13] Danielle Varley,[1] Jonathan Gower[14]

**To cite:** van der Feltz-Cornelis CM, Sweetman J, Edwards M, *et al.* Identifying the top research priorities in medically not yet explained symptoms (MNYES): a James Lind Alliance priority setting partnership. *BMJ Open* 2022;**12**:e061263. doi:10.1136/bmjopen-2022-061263

**Correspondence to**
Professor Christina Maria van der Feltz-Cornelis; christina.vanderfeltz-cornelis@york.ac.uk

## ABSTRACT

**Objectives** This study establishes research priorities for medically not yet explained symptoms (MNYES), also known as persistent physical symptoms or medically unexplained symptoms, from the perspective of patients, caregivers and clinicians, in a priority setting partnership (PSP) following the James Lind Alliance (JLA) approach. Research into such symptoms in general has been poorly funded over the years and so far has been primarily researcher-led with minimal input from patients, caregivers and clinicians; and sometimes has been controversial.

**Design** JLA PSP method. The PSP termed these symptoms MNYES.

**Methods** The study was conducted according to the JLA's detailed methodology for conducting priority setting exercises. It involved five key stages: defining the appropriate term for the conditions under study by the PSP Steering Group; gathering questions on MNYES from patients, caregivers and clinicians in a publicly accessible survey; checking these research questions against existing evidence; interim prioritisation in a second survey; and a final multi-stakeholder consensus meeting to determine the top 10 unanswered research questions using the modified nominal group methodology.

**Results** Over 700 responses from UK patients, caregivers and clinicians were identified in the two surveys and charities contributed from a broad range of medical specialties and primary care. The final top 10 unanswered research questions cover, among others: treatment strategies, personalisation of treatment, collaborative care pathways, training for clinicians and outcomes that matter to patients.

**Interpretation** The top 10 unanswered research questions are expected to generate much needed, relevant and impactful research into MNYES.

## INTRODUCTION

Medically not yet explained symptoms (MNYES), also known as medically unexplained symptoms (MUS) or persistent physical symptoms (PPS),[1] represent up to 30%

### STRENGTHS AND LIMITATIONS OF THIS STUDY

⇒ Establishing research priorities for medically not yet explained symptoms from the perspective of patients, caregivers and clinicians for the first time is a strength of the study.

⇒ The use of the established and transparent James Lind Alliance methodology is a strength of the study.

⇒ Over 700 responses were gathered from patients, caregivers and clinicians from a wide range of medical specialties including primary care, indicating that the priorities were widely supported.

⇒ Contributions of people from ethnic and gender minority groups and from underserved areas within the Priority Setting Partnership Steering Group, surveys and final workshop supports the inclusive nature of this work and indicates these priorities are important to a diverse range of people.

⇒ Self-descriptions of participant roles and symptoms did not always provide sufficient detail to clearly describe the variety of the participants in the sample.

of presentations in primary care and internal medicine settings.[2–4] They can include fatigue, pain, dizziness, irritable bowel syndrome and functional neurological symptoms (FND).[4] They are often deemed to be stress-related, or of psychological origin, and comorbid depressive or anxiety disorder occur in approximately 30% of cases.[3] Patients diagnosed with these symptoms often feel that they are not taken seriously, although care may have been taken to explain their condition properly. It can take a long time to reach the conclusion that patients have MNYES; during this time they typically experience high levels of disability and face repeating appointments and diagnostic procedures. They hear that no cause can be found for their symptoms and this is often delivered by clinicians who have a dualistic view of health and disease. Disability

and absenteeism occur frequently even in patients who present only within primary care with a low number of symptoms and where the effect of demographic factors, anxiety and depressive disorder are taken into account.[5–8] This inevitably leads to disappointment and frustration.[9] Many clinicians lack confidence in the assessment and management of MNYES, or may exhibit behaviours perceived as dismissive. Patients often perceive a stigmatising attitude from clinicians and a sense that they are being judged as neurotic or mentally unwell.[10–12] Moreover, management plans may not be sufficiently holistic to address all patient concerns, and effective treatments are scarce. All the above impact negatively on long-term prognosis.

The focus of research on MNYES is often on particular subsets of symptoms, such as chronic pain, chronic fatigue, irritable bowel syndrome and dizziness, but lacks a comprehensive view. This has ramifications for patients who visit different clinics for their various symptoms, without sustained improvement, and as such experience unmet needs.[13 14] To address this, the University of York through the lead author (CMvdF-C) established a Priority Setting Partnership (PSP) for research needed to address MNYES. We engaged with members of the public, patients with MYNES and their caregivers, clinicians of all medical specialties known to have patients with MNYES,[15] and other key stakeholders such as charities and the Royal College of Psychiatry Liaison Faculty. Close collaboration with the James Lind Alliance (JLA) enabled this PSP to follow their established, rigorous approach to identify the treatment and management priorities of stakeholders (patients, caregivers, clinicians and support organisations) and to incorporate these into a research agenda.[16]

The European Association of Psychosomatic Medicine has published a research agenda in this domain with one of the research priorities being patient preferences for research in this field.[17] Until now, however, there has been relatively little support available for people with MNYES and those who care for them, to enable them in setting up the research agenda. Engaging patients in the research process incorporates their perspective as 'experts' from their unique experience of living with symptoms, as well as their personal knowledge regarding diagnostic trajectories and treatments in the healthcare setting if such symptoms remain (partially) unexplained.[18] This study aims to address this knowledge gap.

The aim of this PSP was therefore to develop a research agenda with the joint perspectives of patients, caregivers, clinicians and support organisations across the UK as the frame of reference, to identify the most important unanswered research questions in MNYES.

## METHODS
This study was undertaken according to the JLA's method for undertaking PSPs as delineated in the JLAs Guidebook.[16] An independent JLA Adviser (JGo) guided the study team through the project and ensured that every step followed the JLA's methodology and adhered to the JLA's principles of transparency and balanced inclusion of patients, caregivers and clinicians. All materials related to this PSP can be found on the JLA website.[19]

### Establishing the Steering Group
In March 2020 the MNYES PSP Steering Group met for the first time. The remit of the Steering Group was to oversee, project manage and publicise the PSP, networking with charitable, patient and professional organisations to maximise the response to the surveys. The Steering Group ensured that the JLA's methodology and principles were adhered to and had no influence on the choice and ranking of the research priorities which were solely determined by the survey responses and final priority setting workshop.

Members of the Steering Group were selected by a snowballing method via clinics and supporting organisations, inviting clinicians providing diagnosis and treatment of the different conditions potentially covered by MNYES. Also, charities, patient networks and PPI networks were approached to recruit patients and caregivers. They were invited and selected based on the capability, motivation and consent to contribute to the JLA PSP working group standards of reference as described in the JLA website. Efforts were made to have a representation of patients with pain, fatigue, FND, IBS and dizziness, as they are the most common MNYES conditions as shown in the literature.[20] Efforts were made to include people from areas outside of London, including rural areas and underserved areas as delineated by clinics and General Practitioners (GPs) in the North of England in the Steering Group. The Steering Group was tasked with overseeing the PSP by having meetings every 6 weeks, chaired by the JLA advisor, and making critical decisions at key points of the project.[19] The composition of the Steering Group is shown in online supplemental box 1.

### Terminology
Many terms are used for these symptoms, including, but not limited to, persistent physical symptoms (PPS),[1] somatic symptom and related disorders,[21] bodily distress disorders,[22] MUS, functional symptoms and functional neurological disorder (FND). There is an ongoing debate among researchers and clinicians about how to refer to these conditions. Many of such terms have been deemed unsatisfactory by patients, caregivers and clinicians as well as researchers for a variety of reasons, leading to ongoing efforts from researchers to find a better term;[12 23 24] however, so far the patient, carer and clinician perspective regarding the choice of preferred term has been lacking. This may seem semantic, however, it underpins the conceptual confusion that exists regarding these symptoms.[25] Unfortunately, in some cases such uncertainty can give rise to deeply rooted controversy that ultimately can be traced back to lack of knowledge regarding the underlying conditions, and to related stigma. This knowledge gap could either be a factual lack of evidence, or a lack

of availability of existing knowledge to clinicians, patients and the general public alike. Therefore, the study's PSP Steering Group took time to decide what terminology to use in the study.

A common concern appeared to be the distress caused to patients, caregivers and clinicians alike by the lack of adequate explanations, diagnostic methods and treatments for these symptoms-which are often poorly understood across these groups too. This was felt to have a negative impact on clinical work and research pertaining to these conditions and to stigmatise them at a societal level. After deliberation, the PSP Steering Group agreed to use the term medically not yet explained symptoms (MNYES) to describe the subject matter for the duration of the study. This was an operational definition not intended to add to or replace other definitions already in use, that was constructed to embrace the views of all stakeholders. MNYES was meant to indicate that although some insights might exist, our understanding is still incomplete. This could pertain to biological, psychological and social factors, as well as factors involving the trajectory of patients through various healthcare settings. In that sense, the choice of the term MNYES conveys a message of hope, which feeds into the effort to identify research priorities for the condition.

### Inclusion and exclusion criteria

The PSP's Steering Group agreed that the remit should include the aetiology, diagnosis and treatment or medical care of patients with MNYES in the UK, as well as the organisation of services, social consequences and long-term outcomes including cost implications for patients. Confirmed topics included (but were not limited to): pain, fatigue, dizziness, FND, bowel symptoms, palpitations and syncope. Ages 16 and older were included. Although fatigue as a symptom was considered for inclusion, chronic fatigue syndrome was considered out of scope since there was another PSP addressing this.

### Patient and public involvement

A core principle of JLA PSPs is collaboration between all stakeholders (patients, caregivers and clinicians) to ensure their views are represented at each stage of the process. At the level of the steering group, patients, caregivers and clinicians were members of the MNYES Steering Group, represented at every meeting, and involved in the development of PSP surveys. They were involved in the organisation of uncertainties, the wording of summary questions, and the verification of evidence checking. At the level of the surveys, patients, public and supporting organisations participated in the surveys as shown in online supplemental table 1. The final workshop also included patients, caregivers and clinicians in the final prioritisation process to establish the top 10 research priorities for MNYES. Furthermore, there were observers representing supporting organisations and relevant charitable organisations during the final workshop. All Steering Group members were invited to contribute to the dissemination of the surveys; the information shared by this PSP was developed with members from all stakeholder groups. All PSP Steering Group members were invited to contribute to the article describing the findings and one of them indeed contributed as a co-author.

## RESULTS

The process is shown in the project flow diagram presented in figure 1.

### First survey

The initial survey (June 2020–January 2021) asked patients, caregivers and healthcare professionals to indicate their priorities for future research related to MNYES.[26] There were 705 respondents who accessed the initial survey; 443 provided at least one question or statement and were included. Included respondents were 77% female, 86% white. Data from the 2011 census show that 51% of the England and Wales population were female[27] and 86% of the same population were white.[28] 68% of the participants were patients or caregivers as reported in online supplemental table 2.

The information specialists (DV and JSw) and PSP lead (CMvdF-C), grouped similar or duplicate questions into five domains, generating 96 draft summary questions on aetiology, diagnosis, healthcare services, treatment, outcomes, prognosis and other. Those 96 draft questions were reviewed by small groups of PSP Steering Group members that comprised clinicians, patients and caregivers. Further consolidations were made resulting in 46 summary questions which were reviewed again and signed off at a meeting of the whole PSP Steering Group. A document illustrating this is available on the JLA website.[29] Of these 46 questions, 22% related to aetiology, 24% to health and clinical services, 15% to diagnosis, 24% to the treatment of MNYES and 15% to outcomes. The proportion of questions posed by stakeholder groups, organised by topic, is shown in online supplemental figure S1.

### Evidence check

The 46 summary questions were checked against published systematic reviews and clinical guidelines. We found that none of the 46 summary questions had been fully answered by previous research; some questions had been answered for specific symptoms, but not comprehensively across all MNYES symptoms. At a subsequent meeting, the Steering Group reviewed the 46 summary questions in relation to the original questions and statements from which they derived. This resulted in minor changes to the wording of these 46 questions which were then included in the interim prioritisation survey.

### Interim survey

This online survey was completed by 270 participants from across the UK. Patients and caregivers made up 74% of the participants. Demographic information is shown in online supplemental table 2.

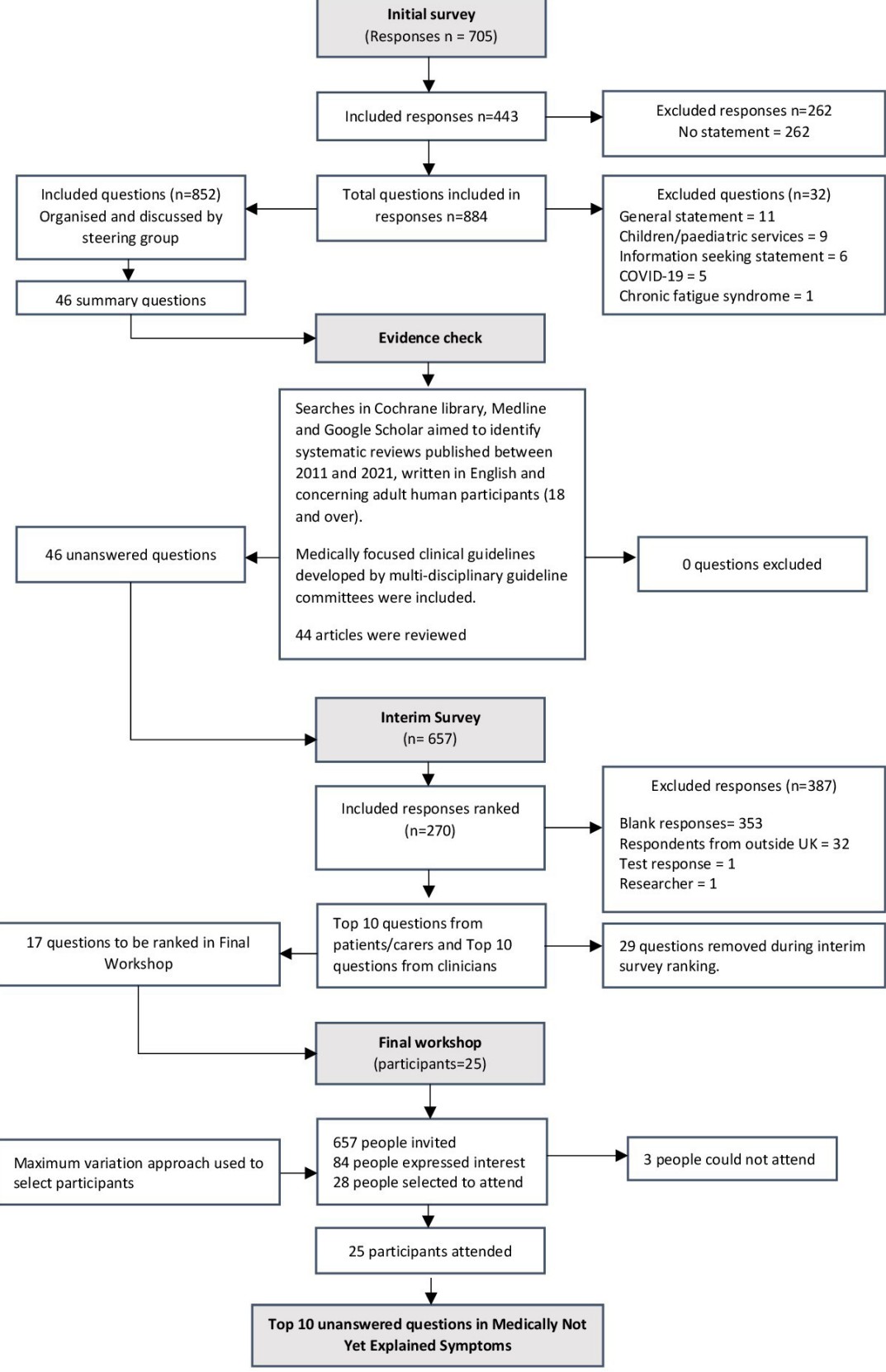

**Figure 1** Flow diagram of medically not yet explained symptoms question prioritisation processes.

## Final priority setting workshop

The final priority setting workshop was conducted remotely over 2 days. In total, 25 people participated in the workshop sessions; 4 JLA advisors facilitated the subgroups, 8 people observed and one person provided technical support. Participants included 11 people

## Top-10 Research Priorities
## Medically Not Yet Explained Symptoms (MNYES)

1. What are the most effective treatment strategies for different symptoms of MNYES?
2. How can clinicians collaborate effectively to form the most appropriate care pathway and service model to offer assessment and treatment for patients with MNYES?
3. What are the most effective methods for training clinicians to diagnose and treat their patients with MNYES with compassion, empathy and respect?
4. What outcomes matter most to patients with MNYES?
5. What are the most effective ways to support patients with MNYES and their carers to live with their symptoms?
6. How can the most appropriate treatment be selected, dependent on different MNYES symptoms, that a person with MNYES is most likely to benefit from?
7. What symptoms are commonly reported by people with MNYES and what links them?
8. What factors affect outcomes for MNYES?
9. What strategies are effective at promoting awareness and up to date clinical knowledge about MNYES amongst healthcare professionals?
10. Which self-management techniques are effective in MNYES?

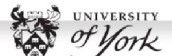 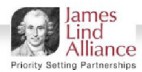 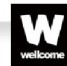 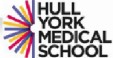

**Figure 2** Top 10 research priorities. MNYES, medically not yet explained symptoms.

with MNYES or caregivers, and 14 healthcare professionals representing psychiatry, general practice, stroke, neurology, physiotherapy, psychology, occupational therapy and gastroenterology. The final top 10 research priorities were agreed by consensus between all the participants as listed in figure 2. They were placed on the JLA website.[30]

The research priorities which were ranked 11–17 are also listed on the JLA website[30] and in online supplemental box 2.

## DISCUSSION
### Summary of the results
In this study, we used the JLA PSP processes to identify the top 10 unanswered research questions for MNYES. We used the collective perspectives of patients, caregivers and clinicians and focused on areas where up-to-date, reliable research evidence is lacking. This process was supported by charitable and professional organisations across the UK. The study highlighted the paucity of evidence-driven practice in MNYES care since none of the 46 research questions gathered from our survey have been previously answered by level I evidence. Based on the extensive discussions during the meetings, the following major themes emerged from the top 10 unanswered research questions.

### Theme 1: treatment
What are the most effective treatment strategies for different symptoms of MNYES?

How can the most appropriate treatment be selected, dependent on different MNYES symptoms, that a person with MNYES is most likely to benefit from?

This pertains to all potential treatment strategies (eg, pharmacological, psychological, physical or collaboration models) to help manage or alleviate any MNYES or combination of symptoms, such as pain, fatigue, dizziness, FND, bowel symptoms, palpitations and syncope. It also focuses on the best ways to match people who experience specific MNYES with the treatments that are most likely to benefit them, personalising treatment based on diagnostic factors such as a history of trauma, biomarkers or treatment needs.

Given the high unmet clinical need and the heterogeneity of MNYES, it is no surprise that this is considered an important research priority. Interestingly, treatment and personalised treatment were also research priorities identified by the Delphi study conducted among experts in the field on behalf of the European Association of Psychosomatic Medicine;[17] they can therefore be considered widely supported research priorities in this field.

### Theme 2: the role of clinicians
How can clinicians collaborate effectively to form the most appropriate care pathway and service model to offer assessment and treatment for patients with MNYES?

What are the most effective methods for training clinicians to diagnose and treat their patients with MNYES with compassion, empathy and respect?

What are the most effective ways to support patients with MNYES and their carers to live with their symptoms?

What strategies are effective at promoting awareness and up to date clinical knowledge about MNYES amongst healthcare professionals?

Four of the ten research priorities involve the role of the clinicians in the diagnostic and treatment process, an indicator of the high relevance of this theme. Many different clinicians provide diagnostic assessments to people with MNYES, or are sought to provide treatment to them. The focus here is on finding the best ways for clinicians to collaborate, forming an appropriate care pathway to support people with MNYES. These could be psychiatric consultation models, multi-disciplinary team models, collaborative care models or other integrated care pathways. There is a focus on communication which acknowledges the perspective and concerns of the person experiencing MNYES. Another priority focuses on identifying options for supporting people with MNYES and their caregivers, such as for example shared decision-making regarding treatment options; coping with symptoms; and rehabilitation approaches. Another priority emphasises strategies to consistently and effectively ensure that clinicians know the most up-to-date information about MNYES and let care reflect current evidence. Given the existing knowledge gaps, this is considered an important priority.

### Theme 3: symptoms and outcomes

What outcomes matter most to patients with MNYES? What symptoms are commonly reported by people with MNYES and what links them?

What factors affect outcomes for MNYES?

Some research priorities mention the patient perspective explicitly. Based on the survey answers, outcomes relevant for patients may include but are not limited to: symptom reduction, changes in biomarkers; improvements in abilities to undertake daily tasks; improvements in quality of life; individual goal achievements or improvements in functioning. The list of MNYES is extensive, and people who experience these symptoms often report living with multiple MNYES. One priority aims to identify the most commonly co-occurring symptoms and their underlying factors and mechanisms. Given the number of questions that pertained to aetiological factors and the fact that the related uncertainty plays a role in the choice of MNYES as a term, this can be considered an important research theme. Factors affecting outcome should include biomarkers, psychological factors, health services, how information is shared between clinicians and people experiencing MNYES, and social factors such as poverty, education, family dysfunction or domestic abuse, trauma and work environment.

### Theme 4: recovery

Which self-management techniques are effective in MNYES?

This priority concerns the strategies that people experiencing MNYES can use separately from clinic visits. The focus is to identify the most effective self-administered therapies for managing or alleviating MNYES, used separately or in combination with formal treatment. Examples of self-management approaches include education,

exercise and dietary changes. It should be noted that this research priority, in contrast to ones covered by the other themes, considers that recovery in MNYES can occur, either by recovery of symptoms or by personal recovery with ongoing symptomatology. Recovery of symptomatology is referred to as clinical recovery and is covered by the other themes. Recovery while symptoms are ongoing is called personal recovery,[31] meaning that despite symptoms being present, the function has to some extent been restored through treatment, self-management or disability management.

In mental health research and clinical practice, especially concerning psychotic conditions, personal recovery is a construct that has increasingly gained attention over the past 30 years; however, the term has not been used in MNYES. Generally, both in clinical practice and in research, the emphasis seems to have been to either attempt to attain clinical recovery or send the patient home with the message that MNYES cannot be cured and that one would have to live with the condition. This dichotomy has fed into the ongoing controversy about how to approach MNYES. This polarising stance is unhelpful. It could provide an essential contribution to further research development in this domain, alongside the research priorities summarised in the other themes. Developing this research priority would require embracing the concept that personal recovery refers to an individual process of adaptation and development where one does not simply return to but instead grows beyond the premorbid self,[32] emphasising the patient perspective.

### Strengths of the study

This is the first study establishing research priorities for MNYES from the perspective of patients, caregivers and clinicians. The study follows the JLA method which offers a unique, and internationally highly regarded, approach to setting research priorities through an equal partnership between patients, carers and healthcare professionals. The priorities represent a 'snapshot in time' of the areas which matter the most to patients, caregivers and clinicians. It is reproducible (the handbook and all relevant materials are available on the JLA website for this purpose) and the method can be used to 'refresh' priorities at a future date to reflect changes in the management of the condition and patient/carer experiences. The use of this established and transparent JLA methodology supports the generalisability of the results and is a strength of this study.

This is a highly contentious area; however, the research priorities were widely supported by over 400 participants including clinicians from a variety of disciplines, patients with a range of symptoms, caregivers, charitible organisations and other supporting organisations. Over 700 responses were gathered from patients, caregivers and clinicians from an array of medical specialties including primary care, indicating that the priorities were widely supported. Contributions of people from ethnic and

gender minority groups and from underserved areas within the PSP Steering Group, surveys and final workshop supports the inclusive nature of this work and indicates these priorities are important to a diverse range of people.

The themes identified in this PSP cover a broad range of ideas, issues and uncertainties; these are outlined in the top 10 research priorities and also reflected in the 7 research priorities that did not make the top 10. Research priorities 11, 12 and 17 would link well with theme 3 in exploring associations of MNYES with mental health and somatic comorbidity, as well as the development of symptoms over time. Priorities 13 and 14 would fit in theme 2, the role of clinicians; 15 and 16 link with theme 1, treatment. This suggests that the themes covered by the top 10 priorities are consistent with the other research priorities which were proposed during this priority setting process.

### Limitations of the study

When comparing the participants of survey 1 with survey 2, there were 443 participants in survey 1, and 270 in survey 2. The final workshop was attended by 25 people. These are high numbers and certainly adequate for priority setting according to the JLA method. However, as the description of the roles is self-described, the variety of investigative participants remains unclear in some respects. For example, it should be pointed out that in online supplemental table 2, 10 people self described as 'doctor', and 8 as 'other' clinician and they may well have been doctors working in primary care as GPs, or rheumatologists; however, we do not know for sure. Regarding the patients, they would state their self-described main symptom as 'pain' in approximately half of the cases; from their answers to the open questions, it emerged that this often would refer to musculoskeletal or rheumatological pain. So, while the exact variety is uncertain, it is unlikely that this contributed to priorities in the final list of issues related to MNYES.

The study provides a good overview of research priorities for MYNES in the UK; however, given the specific cultural aspects and healthcare organisation in the UK, the findings may not be generalisable to other countries. A similar PSP is currently being conducted in the Netherlands and may shed light on research priorities in a non-NHS healthcare setting. This will provide an opportunity to compare and evaluate the generalisability of these findings and the influence of different cultural and healthcare settings. Future research highlighting the situation in low-income and middle-income countries would be beneficial. The results of this PSP will enable funders to prioritise research in MNYES as outlined here and hopefully will provide new, much needed knowledge in this domain.

### CONCLUSION

MYNES are common and reflect a high level of unmet clinical need. Incorporating patient-driven research in

MNYES research can allow researchers to better address the complex care needs of people with MNYES. The most important aspect of this priority setting exercise was strengthening the relationship between patients, caregivers, clinicians and support organisations and generating a list of priorities valued by these stakeholders, which we hope will guide future research.

We have identified the top 10 research priorities in MNYES using the rigorous JLA priority setting method that may open the door to further research addressing the knowledge gaps and controversies in this area, and hopefully alleviate some of the stigma related to these conditions and the people suffering from MNYES. Identification of these top 10 research priorities is an important first step to generating relevant, impactful research that will ultimately improve the lives of people with MNYES.

**Author affiliations**
[1]Department of Health Sciences, University of York, York, UK
[2]Hull York Medical School, University of York, York, UK
[3]York Biomedical Research Institute, University of York, York, UK
[4]R&D Department, Tees Esk and Wear Valleys NHS Foundation Trust, Darlington, UK
[5]Institute of Health Informatics, University College London, London, UK
[6]St George's University of London, London, UK
[7]St George's University Hospitals NHS Foundation Trust, London, UK
[8]Department of Cardiology, University of London Kings College Hospital, London, UK
[9]Liaison team, Tees Esk and Wear Valleys Foundation Trust, York, UK
[10]Patient Representative, North Yorkshire, UK
[11]Department of Endocrinology, Royal Victoria Infirmary, Newcastle upon Tyne, UK
[12]Tranzo Department, Tilburg University, Tilburg, Netherlands
[13]Centre of Excellence for Body Mind and Health, GGz Breburg, Tilburg, The Netherlands
[14]James Lind Alliance, Southampton, UK

**Acknowledgements** The Steering Group would like to thank to all the patients, caregivers, families, friends, healthcare professionals and supporting organisations who contributed to this work. In addition to the coauthors, Philippa Bolton, Sally Brabyn, Tracey Cunningham, Rosie Evans, Miriam Lomas, Margot Metz, Chris Price, Annie Shaw, Scott Spain, Lesley Spain were members of the PSP Steering Group. The following people who attended the final workshop for priority setting agreed to be named: Anna Burneika, Kit Byatt, Phoebe Cole, Tracey Cunningham, Mark Edwards, Rosie Evans, Eve Fazakerley, Jennifer Gilligan, Stephanie Johnston, Claire Jones, Hilary Lewis, Joseph Littlewood, Miriam Lomas, Andrew Moriarty, Elizabeth Paul, Emma Reinhold, Keziah Reiss, James Sampford, Annie Shaw, Gemma Smith, Martin Veysey, Juliet Wilson, Jennifer Wilson.

**Contributors** CMvdF-C was the project lead, instigating the application to the JLA. CMvdF-C, NS, JSw and JGo organised the Steering Group meetings. JSw and DV designed and built the surveys, analysed the data, and conducted the evidence check under supervision of JGo and CMvdF-C. CMvdF-C, JSw and NS wrote the majority of the final manuscript. JGo (JLA Chair) chaired all the meetings, led the consensus workshop and ensured compliance with methodology throughout. NS and JSw took minutes for Steering Group meetings, built and distributed surveys, and organised the consensus workshop. JSw conducted the searches for the evidence check and screened the results with CMvdF-C. Members of the Steering Group CMvdF-C, JSw, ME, NG, JGi, SH, AK, ASM, PP, JSa, NS, IE, DV, PB, SB, TC, RE, ML, MM, CP, AS, SS, LS and JGo (JLA Chair) all attended a majority of the meetings, agreed the initial protocol and the evidence check protocol, piloted and signed off the surveys and disseminated them, checked the raw questions against the indicative ones, reviewed the evidence check results and agreed the final longlist. TC, ME, JGi, ML, ASM, JSw, JSa, AS and JGo (JLA Chair) were present at the final consensus workshop. All authors reviewed and contributed to the final manuscript and approved it prior to submission. CMvdF-C acted as guarantor.

**Funding** This research was funded in whole, or in part, by the Wellcome Trust [Grant number 204829] from the overall Wellcome Trust ISSF award through the Centre for Future Health (CFH) at the University of York. For the purpose of open

access, the author has applied a CC BY public copyright licence to any Author Accepted Manuscript version arising from this submission.

**Competing interests** None declared.

**Patient and public involvement** Patients and/or the public were involved in the design, or conduct, or reporting, or dissemination plans of this research. Refer to the Methods section for further details.

**Patient consent for publication** Not applicable.

**Ethics approval** Not applicable.

**Provenance and peer review** Not commissioned; externally peer reviewed.

**Data availability statement** Data sharing not applicable as no datasets generated and/or analysed for this study. Data relevant to the study are included in the article or on the James Lind Alliance website (https://www.jla.nihr.ac.uk/priority-setting-partnerships/medically-not-yet-explained-symptoms/) and linked University of York website (https://www.york.ac.uk/healthsciences/research/mental-health/projects/mnyes/).

**ORCID iDs**
Christina Maria van der Feltz-Cornelis http://orcid.org/0000-0001-6925-8956
Jennifer Sweetman http://orcid.org/0000-0003-1969-4586
Andrew Stephen Moriarty http://orcid.org/0000-0003-0770-3262

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
