## [Reviewer comments · BMJ Open]

ARTICLE DETAILS

TITLE (PROVISIONAL)	Identifying the top research priorities in medically not yet explained symptoms (MNYES): a James Lind Alliance priority setting partnership
AUTHORS	van der Feltz-Cornelis, Christina; Sweetman, Jennifer; Edwards, Mark; Gall, Nicholas; Gilligan, Jennifer; Hayle, Steph; Kaul, Arvind; Moriarty, Andrew; Perros, Petros; Sampford, James; Smith, Natalie; Elfeddali, Iman; Varley, Danielle; Gower, Jonathan

VERSION 1 – REVIEW

REVIEWER	Buszewicz, Marta University College London, Research Department of Primary Care and Population Health
REVIEW RETURNED	06-Mar-2022

GENERAL COMMENTS	I think this paper aims to address an important subject area and am aware that the views of patients tend to be under-represented in much current research in this field. I didn't however find it a very easy paper to read and in particular think insufficient details were given for the study to be repeated by any other researchers. This is despite it being a long paper as presented, but maybe some of the text in the Introduction and Methods could be made more concise to allow for a clearer description of the methods used. I have the following comments and suggestions: (1) I wasn't really convinced about the introduction of yet another term - MNYES to describe this problem and wonder if it would make it difficult in future for this paper to be identified in literature searches given I don't think it's currently in general usage. Is the idea that the patients in the group called MNYES are the same as those with MUS or different in some way - the way the literature is referred to would imply that they're considered the same. (2) I appreciate that this term was decided upon by the Steering Group but I was concerned about the balance of the group, given it seemed to involve 10 clinicians from various relevant specialties but only 4 patients and 1 caregiver, so wasn't really in a position to give strong representation to the patient voice which had been highlighted as being very important. I was unclear who the information specialists and project coordinators were and their role on the Steering Group - i.e. were they there to facilitate or would they be involved in the discussion and decision making. (3) I was concerned in the first paragraph of the Introduction about how the patient group was being described. Given it was
--

	acknowledged that MUS or MNYES represent up to 30% of presentations in primary care and internal medicine, I think it's important to acknowledge there is a whole spectrum of severity of presentations and by no means all or even the majority of patients presenting in primary care or general medical settings will have significant functional disability or require specialist input - they may be able to be managed within a primary care or general medical context providing the relevant clinicians have the appropriate skills. (4) As such I was uncertain what criteria were used to identify the patients invited to be on the Steering Group as well as those patients invited to take part in the two surveys and final workshop and whether any objective criteria were used. I was also unclear how these patients were selected and invited - I couldn't find any clear details given about this. The use of partner organisations web-sites was mentioned but I was unsure if this meant all the supporting organisations listed in Table 1? The reason why these particular organisations were selected was unclear and didn't seem to cover the whole potential range of MYNES - e.g. there was no reference to any cardiology organisation. I was unsure why the British Thyroid association was included, as I'm not aware of thyroid problems often being MYNES. I wasn't previously aware of the Graham Hughes International organisation - on looking this up it appears to focus on anti-phospholipid syndrome and I'm unsure how this connects with MYNES/MUS. (5) I was unclear what the process was in moving from the initial survey to the interim survey (initially asking for suggestions about unanswered questions and then ranking these) and whether the same people were approached each time, given that the numbers in the interim survey appeared to be overall significantly lower, but higher in a few categories such as the GPs. (6) I was also unclear how the clinicians listed in Table 2 had been selected and invited to take part? I was struck by the large number of psychiatrists involved in the initial survey (at 44 this was around the same number as all the other clinicians combined) and what this reflected, given I'm not aware of psychiatrists playing such an overtly major role in the management of such patients. This contrasted with only 4 GPs being involved, despite the large number of patients presenting in primary care. There didn't appear to be any rheumatologists involved, despite this being a medical specialty often involved in the care of such patients and no participating patients were described as having musculo-skeletal or rheumatological symptoms which appears to be an omission. (7) I think the Top 10 research priorities as given in Figure 3 are clearly important, but each one would form a very major programme of work. There have already been a great many studies carried out in this field and I think it would be helpful to initially examine the existing literature to see how this could be used to answer some if not all of the questions raised. (8) In terms of limitations of the study I would include some of my points raised above which would make it very difficult to replicate this work without further information. I also think there are significant limitations in terms of lack of clarity as to how the patient representatives were selected, which makes it unclear
--	---

	whether they could be considered representative of the MNYES population as a whole. (9) I found the Discussion very long and complex to read in terms of clarity and message. I would suggest it could be made much more concise, summarising the important points for each research question within each theme.
--	--

REVIEWER	Kathol, Roger University of Minnesota Twin Cities, Psychiatry
REVIEW RETURNED	15-Mar-2022

GENERAL COMMENTS	The findings from the study are clear. While my personal feeling is that the article could be shorter and still understood, journal publication requirements would make this difficult. The authors do a good job of including critical contributors to the understanding of MNYES (patients, carers, and clinicians), which provides a better understanding of priorities for MNYES patients and assessing clinicians. It is uncertain why the variety of investigative participants in the final stage was not more broad-based, however, it is unlikely that this contributed to priorities in the final list of issues related to MNYES.
---

VERSION 1 – AUTHOR RESPONSE

Reviewer: 1

Dr. Marta Buszewicz, University College London

Comments to the Author:

I think this paper aims to address an important subject area and am aware that the views of patients tend to be under-represented in much current research in this field.

We appreciate the acknowledgement of the importance to cover this subject by the reviewer.

I didn't however find it a very easy paper to read and in particular think insufficient details were given for the study to be repeated by any other researchers. This is despite it being a long paper as presented, but maybe some of the text in the Introduction and Methods could be made more concise to allow for a clearer description of the methods used. I have the following comments and suggestions:

We moved some parts of the methods section to a supplementary file to shorten the paper. We followed the suggestions of the reviewer below that were intended to clarify the paper.

(1) I wasn't really convinced about the introduction of yet another term - MNYES to describe this problem and wonder if it would make it difficult in future for this paper to be identified in literature searches given I don't think it's currently in general usage. Is the idea that the patients in the group called MNYES are the same as those with MUS or different in some way - the way the literature is referred to would imply that they're considered the same.

This is correct, MNYES are considered the same as MUS, PPS and the other terms that are mentioned in box 1. We now state this early in the abstract and in the introduction as follows:

Medically Not Yet Explained Symptoms (MNYES), also known as Medically Unexplained Symptoms (MUS) or Persistent Physical Symptoms (PPS),...

In addition, to facilitate finding this study in literature searches, as MUS was already a keyword for the study, we now added PPS to the key words as well.

(2) I appreciate that this term was decided upon by the Steering Group but I was concerned about the balance of the group, given it seemed to involve 10 clinicians from various relevant specialties but only 4 patients and 1 caregiver, so wasn't really in a position to give strong representation to the patient voice which had been highlighted as being very important.

The membership of the Steering Group was constituted according to the protocol set out in the JLA's Guidebook for PSPs:

"The Steering Group is made up of key organisations and individuals who collectively can represent all or most issues related to the Priority Setting Partnership (PSP), either individually or through their networks."

The four patient representatives (with a wide variety of symptoms) and one carer provided an appropriate balance with the healthcare professionals who were in larger numbers because they represented ten clinical specialties (given the wide ranging nature of MNYES). All Steering Group meetings were chaired by the independent JLA Adviser who ensured that the views of the patients were given equal weight to those of the professional members. We revised the description of the setting up of the Steering Group to cover this issue in the Methods section as follows:

This study was undertaken according to the JLA's method

for undertaking PSPs as delineated in the JLA's Guidebook (11). An independent JLA Adviser (JG) guided the study team through the project and ensured that every step followed the JLA's methodology and adhered to

the JLA's principles of transparency and balanced inclusion of patients, caregivers and clinicians. All materials related to this PSP can be found on the JLA website.

and:

The members of the Steering Group were selected by a snowballing method, asking clinics and clinicians providing diagnosis and treatment of the different conditions potentially covered by MNYES to signpost colleagues with an interest in MNYES. Also, charities, patient networks and PPI networks were approached to identify patients and caregivers. They were invited and selected based upon

their capability, motivation and consent to contribute to the JLA

PSP Steering Group standards of reference as described on the JLA website. Efforts were made to have a representation of patients with pain, fatigue, FND, IBS and dizziness, as they are the most common MNYES conditions as shown in the literature. Also, efforts were made to include people from areas outside of London, including rural areas and underserved areas in the North of England in the Steering group.

In addition, after the launch of survey one and two, approximately 700 statements were made about this topic to support the priority setting, all of which can be found on the JLA website.^[1] None of them were specifically critical of the term MNYES. Only two statements mentioned it in their comments as follows:

- Can we please have a more accurate definition of MNYES? In order to formulate scientifically answerable questions, it is imperative that we have a robust definition.
- Is there a better name to describe this problem than MNYES

We feel that this supports

that the term MNYES, after being suggested by the PSP Steering Group that followed JLA guidelines, was generally accepted in the surveys and by the supporting charities, networks and other stakeholders mentioned in Table 1 who distributed the surveys.

I was unclear who the information specialists and project coordinators were and their role on the Steering Group - i.e. were they there to facilitate or would they be involved in the discussion and decision making.

We now make clear in Box 1 that they facilitated and did not engage in the priority setting as follows:

...to facilitate the PSP Steering Group. They prepared meeting documents, surveys, supported recruitment, completed evidence checking and analysis but did not engage in the priority setting.

(3) I was concerned in the first paragraph of the Introduction about how the patient group was being described. Given it was acknowledged that MUS or MNYES represent up to 30% of presentations in primary care and internal medicine, I think it's important to acknowledge there is a whole spectrum of severity of presentations and by no means all or even the majority of patients presenting in primary care or general medical settings will have significant functional disability or require specialist input - they may be able to be managed within a primary care or general medical context providing the relevant clinicians have the appropriate skills.

We agree that there is a whole spectrum of severity of presentations of MNYES in primary care. However, as the work of many researchers in this domain shows, disability occurs in such symptoms in primary care even if severity is on the less severe end in terms of number of symptoms or comorbidity. We now added the following to the introduction to point this out:

Disability and absenteeism occurs frequently even in patients who present only within primary care with a low number of symptoms and where the effect of demographic factors, anxiety and depressive disorder are taken into account.

(4) As such I was uncertain what criteria were used to identify the patients invited to be on the Steering Group as well as those patients invited to take part in the two surveys and final workshop and whether any objective criteria were used. I was also unclear how these patients were selected and invited - I couldn't find any clear details given about this.

Patients and carers were identified and invited

via the PPI networks and clinics of the participating Steering Group members and were selected based upon

the capability, motivation and consent to contribute to the JLA

PSP Steering Group standards of reference as described on the JLA website. Efforts were made to have a representation of patients with pain, fatigue, FND, IBS and dizziness as they are the most common MNYES conditions as shown in the literature. We apologise for the lack of clarity and now added this to the Methods section as follows:

The members of the Steering Group were selected by a snowballing method, asking clinics and clinicians providing diagnosis and treatment of the different conditions potentially covered by MNYES to signpost colleagues with an interest in MNYES. Also, charities, patient networks and PPI networks were approached to identify patients and carers. They were invited and selected based upon the capability, motivation and consent to contribute to the JLA PSP Steering group standards of reference as described in the JLA website. Efforts were made to have a representation of patients with pain, fatigue, FND, IBS and dizziness, as they are the most common MNYES conditions as shown in the literature. Also, efforts were made to include people from areas outside of London, including rural areas and underserved areas as delineated by clinics and GPs in the North of England in the Steering Group.

The use of partner organisations web-sites was mentioned but I was unsure if this meant all the supporting organisations listed in Table 1?

Yes, this is the case, as indicated in the text.

The reason why these particular organisations were selected was unclear and didn't seem to cover the whole potential range of MNYES - e.g. there was no reference to any cardiology organisation.

The cardiologist member of the PSP Steering group recommended POTS-UK to support distribution, as POTS is often the main condition in patients with unexplained symptoms in the cardiological domain, and this organisation indeed supported the dissemination of the surveys.

I was unsure why the British Thyroid association was included, as I'm not aware of thyroid problems often being MNYES.

The endocrinologist member

of the PSP Steering group recommended *the British Thyroid Association* to support distribution, as pa

tients with unexplained symptoms often visit endocrinologists to have thyroid function checked and may have subclinical forms that are subsequently considered functional.

I wasn't previously aware of the Graham Hughes International organisation - on looking this up it appears to focus on anti-phospholipid syndrome and I'm unsure how this connects with MYNES/MUS.

We understand the question. Patients with MNYES often end up being in contact with associations covering very rare conditions and we wanted to reach such patients as well. The fibromyalgia/rheumatological disorders member

of the PSP Steering group recommended this association to support distribution, as patients with unexplained symptoms often end up having such conditions checked.

We mentioned in the text accordingly: **Some of them collaborated because they found that patients with unexplained symptoms often visit their websites and related specialists to assess their symptoms.**

(5) I was unclear what the process was in moving from the initial survey to the interim survey (initially asking for suggestions about unanswered questions and then ranking these) and whether the same people were approached each time, given that the numbers in the interim survey appeared to be overall significantly lower, but higher in a few categories such as the GPs.

The surveys were performed independently of each other. The surveys being conducted anonymously did not allow for reapproaching people after the first survey. We now added this sentence:

The second survey was launched independently from the first survey.

(6) I was also unclear how the clinicians listed in Table 2 had been selected and invited to take part? I was struck by the large number of psychiatrists involved in the initial survey (at 44 this was around the same number as all the other clinicians combined) and what this reflected, given I'm not aware of psychiatrists playing such an overtly major role in the management of such patients. This contrasted with only 4 GPs being involved, despite the large number of patients presenting in primary care.

It should be pointed out that in Table 2, 10 people self described as "doctor," and 8 as "other" clinician and they may well have been doctors working in primary care as GPs.

The clinicians had been invited in a similar manner as the patients, as mentioned in the text, i.e. via social media, facebook groups, websites, via relevant clinics and primary care practices. Some associations supported the distribution of the survey, such as the Faculty of Liaison

Psychiatry, as many psychiatrists providing management of such conditions are consultation liaison psychiatrists. The Royal College of General Practitioners was

approached with the request to distribute the two surveys but declined, stating that non-distribution of surveys was their general approach. Then several members

of the PSP Steering group reached out to GPs and primary care practices known to them, and several supporting organisations approached primary care practices or networks known to them, to offer the survey link. Given the numbers of GPs reached this way, this should have provided at least as many GP contributions as via the RCGP. The timing of the project during the pandemic may have played a role in rather low uptake by GPs, who are known to have been generally overwhelmed with clinical work at that time.

VERSION 2 – REVIEW

REVIEWER	Buszewicz, Marta University College London, Research Department of Primary Care and Population Health
REVIEW RETURNED	06-Jun-2022
GENERAL COMMENTS	I think the authors have answered most of the points raised in my initial review satisfactorily.

	I would however prefer it if they could include a statement to the effect that their method of recruiting patients, clinicians and special interest groups for the Steering Group and surveys via personal recommendation and snowballing could be open to some bias and subjectivity in the opinions given and is a potential limitation for the generalisability of the findings. It would also be difficult for the study to be repeated given the details given. I was disappointed to see the authors did not appear to take any account of my comment (9) that the Discussion was very long and complex to read in terms of clarity and message. This remains my opinion and I think it could be quite effectively shortened without losing the main points which the authors would like to make. I will however leave that to the Journal Editors to decide.
--	---

VERSION 2 – AUTHOR RESPONSE

Reviewer 1:

I think the authors have answered most of the points raised in my initial review satisfactorily.

We are pleased to read that the reviewer was satisfied with most of our answers.

I would however prefer it if they could include a statement to the effect that their method of recruiting patients, clinicians and special interest groups for the Steering Group and surveys via personal recommendation and snowballing could be open to some bias and subjectivity in the opinions given and is a potential limitation for the generalisability of the findings.

We would like to point out to the reviewer that our method did not involve selection of participants based upon personal recommendation. "Personal recommendation" is mentioned nowhere in our manuscript. It is not mentioned anywhere in the Handbook of the James Lind Alliance, the established method that we followed meticulously to achieve the highest possible level of transparency and generalisability. The membership of the Steering Group was constituted according to the protocol set out in the JLA's Guidebook for PSPs: "The Steering Group is made up of key organisations and individuals who collectively can represent all or most issues related to the Priority Setting Partnership (PSP), either individually or through their networks."

In order to avoid any possible misunderstandings regarding this point, we adapted the text regarding the recruitment as follows:

Members of the Steering Group were selected by a snowballing method via clinics and supporting organisations, inviting clinicians providing diagnosis and treatment of the different conditions potentially covered by MNYES. Also, charities, patient networks and PPI networks were approached to recruit patients and caregivers.

The recruitment process may have been subject to some bias, as is the case with all studies conducted ba

sed upon a

Delphi process involving patients, carers and clinicians. In such research, the recommendations can only be based on the responses and self-reported symptoms of those who were keen to be involved. Such bias was mitigated as far as possible by the well trodden James Lind Alliance partnership methodology.

Also, it should be kept in mind that the members of the steering group had no influence on the survey responses and the results. They were supported by the JLA methodology, the chair and the moderators to follow the input from the participants in the surveys and the final workshop. To clarify this further, we adapted the text as follows:

In March 2020 the MNYES PSP Steering Group met for the first time. The remit of the Steering Group was to oversee, project manage and publicise the PSP, networking with charitable, patient and professional organisations to maximise the response to the surveys. The Steering Group ensured that the JLA's methodology and principles were adhered to and had no influence on the choice and ranking of the research priorities which were solely determined by the survey responses and final priority setting workshop.

It would also be difficult for the study to be repeated given the details given.

The study follows the JLA method and is therefore reproducible. For example, the JLA is currently undertaking a "refreshing" of PSP priorities from PSP's that reported several years ago, following their methodology for that. The Handbook and all relevant materials are available on the JLA website or the website of the University of York for this purpose. We adapted the text to make this explicit in the discussion as follows:

The study follows the JLA method which offers a unique, and internationally highly regarded, approach to setting research priorities through an equal partnership between patients, carers and healthcare professionals. The priorities represent a "snapshot in time" of the areas which matter the most to patient, caregivers and clinicians. It is reproducible (the Handbook and all relevant materials are available on the JLA website for this purpose) and the method can be used to "refresh" priorities at a future date to reflect changes in the management of the condition and patient/carer experiences. The use of this established and transparent James Lind Alliance methodology supports the generalisability of the results and is a strength of this study.

I was disappointed to see the authors did not appear to take any account of my comment (9) that the Discussion was very long and complex to read in terms of clarity and message. This remains my opinion and I think it could be quite effectively shortened without losing the main points which the authors would like to make. I will however leave that to the Journal Editors to decide.

We edited the discussion to make it shorter and more palatable to the reader. We also had to extend the discussion in order to address questions of the reviewer regarding, amongst others, the variety of participants, symptoms and disorders covered, and generalisability of the findings. We were therefore able to reduce the discussion section with 161 words.